# Construction of a standardized training system for hospital infection prevention and control for new medical staff in internal medicine ICUs based on the Delphi method

Linfei Wu[1], Li Tang [1]*, Linli Zhuang[2], Wenyi Xie[1], Min Liu[3], Jianfang Li[1]

**1** West China School of Nursing, Sichuan University /Department of Medical Intensive Care Unit, West China Hospital, Sichuan University, Chengdu, China, **2** Department of Rheumatology and Immunology, West China Hospital, Sichuan University, Chengdu, China, **3** Department of Center for Infectious Diseases, West China Hospital, Sichuan University, Chengdu, China

\* 18980601909@163.com

**Data Availability Statement:** All relevant data are within the manuscript and its Supporting Information files.

## Abstract

In China, studies have shown nosocomial infections contribute to increased mortality rates, prolonged hospital stays, and added financial burdens for patients. Previous studies have demonstrated that effective infection control training can enhance the quality of infection control practices, particularly in intensive care unit (ICU) settings. However, there is currently no universally accepted training mode or program that adequately addresses the specific needs of ICU medical staff regarding nosocomial infection control. The objective of this study was to develop a standardized training system for preventing and controlling hospital-acquired infections among new medical staff in the internal medicine ICU. Our methodology encompassed an extensive literature review, technical interviews focusing on key events, semi-structured in-depth interviews, and two rounds of Delphi expert correspondence. We employed intentional sampling to select 16 experts for the Delphi expert consultation. Indicators were chosen based on an average importance score of >3.5 and a coefficient of variation of <0.25. The weight of each indicator was determined using the analytic hierarchy process. The efficacy of the two rounds of questionnaires was also evaluated. Our findings revealed that the questionnaires achieved a 100% effective recovery rate, with expert authority coefficients of 0.96 and 0.90. The Kendall coordination coefficients for the first-, second-, and third-level indicators in the initial round of expert consultation questionnaires were 0.440, 0.204, and 0.386 (P < 0.001), respectively. In the second round of expert consultation questionnaires, the Kendall coordination coefficients for the first, second, and third-level indicators were 0.562, 0.467, and 0.556 (P < 0.001), respectively. The final training model consisted of four first-level indicators (hospital infection prevention and control training content, training methods/forms, assessment content, and evaluation indicators), 26 second-level indicators, and 44 third-level indicators. In conclusion, the proposed standardized training system for infection prevention and control among new medical staff in the internal medicine ICU is both scientifically sound and practical, which can contribute to

**Funding:** 1.Initials of the authors who received each award:Wu Linfei 2.Grant numbers awarded to each author:HXHL21063 3.The full name of each funder:West China Nursing Discipline Development Special Fund Project, Sichuan University(NO. HXHL21063) 4.URL of each funder website:N/A 5. Did the sponsors or funders play any role in the study design, data collection and analysis, decision to publish, or preparation of the manuscript?:The funders had no role in study design, data collection and analysis, decision to publish, or preparation of the manuscript.

**Competing interests:** The authors have declared that no competing interests exist.

improved patient safety, reduced healthcare costs, and enhanced overall quality of care in internal medicine ICUs. Moreover, it can serve as a framework for future training projects.

## Introduction

The intensive care unit (ICU) is a specialized ward for critically ill patients and is particularly prone to hospital-acquired infections [1, 2]. Studies have revealed that the incidence of hospital infections in ICU patients ranges from 20% to 60% [3], which is five to ten times higher than that in general hospital departments [4]. ICU-acquired infections not only lead to extended hospital stays and increased financial burdens for patients but also result in higher mortality rates, making them a significant cause of death among ICU patients. Effective prevention measures have shown the potential to prevent more than 33% of nosocomial infections [5]. Consequently, the prevention and control of nosocomial infections in the ICU hold great importance in reducing the risk of infection among ICU patients.

The ICU primarily serves critically ill patients who require invasive diagnostic and treatment procedures, leading to a heightened risk of infection [6]. Despite the existence of standardized protocols for invasive procedures in the ICU, such as various types of intubation, adherence to these protocols is not always consistent among staff members. Previous studies have highlighted suboptimal rates of hand hygiene compliance among ICU medical staff, inadequate adherence to maximum sterile barrier precautions for central venous catheters, and the need for improved training, supervision, and feedback [7–9]. Consequently, experts have suggested that strengthening training in nosocomial infection knowledge is necessary to enhance in-hospital infection management [10].

A meta-analysis of 43 studies examining different training methods and their effects on nosocomial infection knowledge revealed that training programs for medical staff increased knowledge awareness by 41.80% [11]. The training targeted various personnel, including on-the-job medical staff, interns, new employees, and cleaning staff. The training methods employed included theoretical lectures, theoretical learning combined with demonstration operations, and comprehensive approaches. Liu Ying [12] proposed a multi-level, sub-specialty, and sub-form training concept for infection control personnel, demonstrating that the implementation of this diversified training model led to an increase in hand hygiene compliance from 86.38% to 92.27% and a decrease in infection rates from 2.04% to 0.63%. Wang Peng [13] discussed the effects of face-to-face teaching, online training, entrance training, and knowledge competitions on enhancing medical staff's understanding of hospital infection-related knowledge. The findings revealed that while online learning was the most popular, its effectiveness was inferior to the other three methods. The possible reasons for this disparity may be attributed to the flexibility and convenience of online training, which lacks real-time supervision and may result in incomplete implementation. Consequently, the researchers suggested that a comprehensive training approach incorporating face-to-face lectures, online training, entrance training, and knowledge competitions should be considered. Additionally, research has explored other training methods, such as question teaching, case teaching, and feedback teaching, and role-playing. In summary, although these studies have been conducted in China, there remains a lack of a systematic and standardized training and evaluation system for hospital infection prevention and control. Specifically, there is a dearth of research on standardized training approaches for nosocomial infection prevention and control targeting newly recruited medical staff in the ICU.

Research has highlighted the significance of standardized training in nosocomial infection prevention and control as an effective approach to influence the knowledge, attitudes, and behaviors of individuals and groups [14, 15]. However, the existing training methods do not adequately address the specific requirements of ICU nosocomial infection prevention and control due to the unique characteristics of clinical work in the ICU and the diverse personnel involved. Furthermore, the current training approaches lack a standardized training model. Therefore, the objective of this study was to develop a standardized training system for hospital infection prevention and control targeting newly recruited medical staff in the internal medicine ICU. This system aims to improve the awareness of hospital infection prevention and control among medical staff and serve as a reference for establishing a comprehensive standardized training framework for hospital infection prevention and control.

## Materials and methods

### Study design

To carry out this study, we formed a research team comprising an associate professor, two advanced practice nurses, and two graduate students. Their responsibilities included conducting a thorough literature review, selecting experts, distributing and collecting letter questionnaires, and organizing and analyzing recommendations and data.

Initially, a qualitative literature search was conducted to compile a list of potential items for standardizing the training system of hospital infection prevention and control for newly recruited medical staff in the internal medicine ICU. After reviewing the abstracts of the retrieved literature, we identified eligible studies and extracted frequently recurring items relevant to this study. These recorded items were then discussed with the expert group, resulting in a pool of candidate items for constructing the standardized training system. To obtain expert opinions on the standardized training system, the Delphi method was employed, conducting consultations with experts specializing in hospital infection prevention and control for newly admitted medical staff in the internal medicine ICU between April and June 2022.

The Delphi research method involves anonymous feedback from experts who are unaware of each other's identities and are unable to engage in direct discussions or exchanges of opinions during the research process. Multiple rounds of questionnaires are utilized to seek expert opinions [16], with no deliberations or exchanges of ideas [17]. Consensus is considered to have been reached when at least 70% of the experts agree on a particular opinion [18]. Therefore, this approach is deemed effective for this study.

### Participants

The Delphi method involves a varying number of experts, ranging from 5 to 20, 15 to 60, or even exceeding 1,000 [17]. The selection of experts is a crucial factor in the success of the Delphi method. Generally, it is necessary to choose experts who possess expertise in the field of study, possess relevant experience, and demonstrate a keen interest in research [19]. For this study, a total of 16 experts were selected using intentional sampling. In the first round, 16 questionnaires were distributed, and all were successfully completed and returned, resulting in a response rate of 100%. In the second round, a questionnaire survey was conducted among the 16 experts, with a 100% response rate achieved. The selected experts represented two provinces and cities, namely Sichuan and Fujian Province (China). The criteria for expert selection included the following: (1) possessing a bachelor's degree or higher; (2) holding an intermediate title or higher; (3) being experts in hospital infection management and nursing with prior experience in hospital infection prevention and control; and (4) volunteering to participate in the study.

## Data collection

In this study, we developed an interview outline by initially referring to the Knowledge, Belief, and Action Model (KAPB) through a literature review. This outline was further refined based on the research objectives and group discussions involving experienced medical staff engaged in clinical teaching and new ICU medical staff. The selection criteria for medical staff involved in clinical teaching included: (1) being nursing team leaders or doctors with clinical teaching responsibilities, (2) having a minimum of 10 years of experience in ICU work, and (3) expressing a willingness to share their insights and recommendations regarding hospital infection prevention and control competencies for clinical teaching and new medical staff. The selection criteria for new ICU medical personnel were: (1) being newly appointed doctors, nurses, or technicians in the medical ICU (MICU), (2) holding a professional qualification certificate, and (3) expressing a willingness to share their experiences and perspectives on ICU hospital infection prevention and control. In March 2022, we conducted semi-structured interviews with two nursing team leaders and three new nurses in the internal medicine ICU. Prior to the interviews, informed consent was obtained from all participants, and audio recordings were transcribed into written text within 48 h of the interviews.

The data collected was organized and analyzed using Nvivo 12.0 software, and an inductive content analysis approach was employed to identify elements of competency [20]. Ultimately, a preliminary model comprising four first-level indicators, twelve second-level indicators, and sixty-seven third-level indicators was constructed based on a combination of literature review findings and qualitative research. This model served as an evaluation index system for assessing competencies.

## Expert consultation

**Expert letter questionnaire.**   The expert letter questionnaire consisted of four sections: (1) Research introduction, which provided the background, purpose, and significance of the study, along with instructions for completing the questionnaire and details about the researcher's identity; (2) Basic information of the experts, including age, gender, education, professional title, and professional field; (3) Construction of an expert consultation form, where the importance of indicators was assessed using the Likert five-point scoring method. The scale ranged from "very important" (5 points) to "not significant" (1 point). Experts were also given the opportunity to suggest additions, deletions, or modifications to the indicators; and (4) Experts were asked to provide their judgment basis and indicate their familiarity with the research content and indicators.

**Implementation expert consultation.**   Initially, the letter questionnaires were distributed to experts via email or WeChat (Tencent, Shenzhen, China), following the acquisition of informed consent. Simultaneously, experts were requested to rate and provide revisions for each item. After the completion of the first round of questionnaire collection, the research team collated and analyzed the expert opinions. Indicators were adjusted based on a mean value > 3.5 and a coefficient of variation < 0.25. Following the revisions, the second round of expert consultation was conducted, and the results were thoroughly discussed and analyzed. This process ultimately led to the finalization of indicators for the preliminary model.

**Ethics.**   This study adhered to the principles outlined in the Declaration of Helsinki and received approval from the Ethics Committee of West China Hospital, Sichuan University (Approval No. 2022 Annual Review (232); approved on March 9, 2022). Prior to the investigation, the purpose of the study was communicated to the experts, and their verbal consent was obtained. Participants were informed that they could withdraw from the survey at any point during the data collection process.

**Quality management.** To ensure the reliability of the study findings, the selection of experts was standardized using the inclusion criteria mentioned previously. A total of 16 experts from Sichuan and Fujian Provinces and cities were chosen. The research team summarized and analyzed the expert opinions. Kendall's coordination coefficient and the chi-square value were employed during the data analysis process.

**Analytic hierarchy process.** The subjective judgments of the experts were subjected to quantitative scaling using the analytic hierarchy process [21]. The Saaty scale was determined based on the mean difference in importance assignments of evaluation indicators in the second round of expert consultation, and a hierarchical structure model was established. A judgment matrix was constructed, and single-level ranking and consistency checks were performed to determine the weight and composite weight of each index. A consistency ratio below 0.1 indicates a reasonable weight distribution for each indicator and good consistency [22].

## Statistical analysis

Data analysis was conducted using the Statistical Package for the Social Sciences (SPSS, version 25.0; IBM, Armonk, US) software. Descriptive statistics such as mean and standard deviation were used to describe measurement data, while count data were described using frequency and percentage. The effectiveness of expert participation was measured by the questionnaire's effective recovery rate, and the authority of the experts was expressed through the average value of the judgment coefficient and familiarity coefficient. The level of agreement among expert opinions was assessed using the Kendall coordination coefficient, which ranges between 0 (disagree) and 1 (completely agree). Statistical significance was determined at $P<0.05$. The weights of the indicators were calculated using the analytic hierarchy process.

## Results

### Basic information of experts

Following the selection criteria for experts, intentional sampling was employed in this study to select 16 specialists from Sichuan and Fujian Provinces and cities. All experts possessed experience in hospital infection management and training. The experts had a mean age of 42.87 ±5.78 years, with an average of 12.07 ±4.61 years of work experience. Table 1 presents the demographic characteristics of the experts.

### Study participation

The level of engagement among the experts was evaluated by assessing the response rate of the questionnaire. In the first round, all 16 questionnaires distributed were successfully recovered, resulting in a recovery rate of 100%. Likewise, in the second round, all 16 questionnaires distributed were retrieved, achieving a recovery rate of 100%.

### Expert authority coefficient and opinion coordination degree

The expert authority coefficient measures the level of expertise and influence of each expert, while the opinion coordination degree quantifies the degree of agreement among experts. In the initial and subsequent rounds of expert consultation, the authority coefficients were determined to be 0.92 and 0.96, respectively, surpassing the threshold of an expert consultation authority coefficient > 0.7 [17]. The Kendall coordination coefficients for the first-, second-, and third-level indicators in the first round were 0.440, 0.204, and 0.386, respectively (Table 2). In the second round of expert consultation, the Kendall coordination coefficients

**Table 1. Demographic information of experts.**

| Categories | Project | Frequency (N) | Proportion (%) |
|---|---|---|---|
| **Gander** | Male | 3 | 18.75 |
| | Female | 13 | 81.25 |
| **Age (years)** | 30–39 | 5 | 31.25 |
| | 40–49 | 10 | 62.50 |
| | ≥50 | 1 | 6.25 |
| **Work years** | 3–9 | 3 | 18.75 |
| | 10–19 | 10 | 62.50 |
| | 20–29 | 3 | 18.75 |
| | ≥30 | 0 | 0 |
| **Title** | Intermediate level | 9 | 56.25 |
| | Associate Senior level | 4 | 25 |
| | Senior level | 3 | 18.75 |
| **Degree** | Undergraduate | 7 | 43.75 |
| | Master | 6 | 37.5 |
| | Doctor | 3 | 18.75 |
| **Research field** | Infection Prevention and Control | 7 | 43.75 |
| | Clinical research in intensive care unit | 9 | 56.25 |
| **Department** | Infection Prevention and Management | 7 | 43.75 |
| | ICU | 9 | 56.25 |
| **Is it a postgraduate supervisors** | Yes | 3 | 18.75 |
| | No | 13 | 81.25 |

for the first-, second-, and third-level indicators were 0.562, 0.467, and 0.556, respectively (Table 3). All Kendall tests yielded statistically significant results (all P<0.001).

## Training model of hospital infection prevention and control for new ICU medical staff

The Delphi method was employed to conduct two rounds of consultation. In the first round, the research team incorporated feedback from experts, along with the exclusion criteria, and engaged in discussions to refine the feasibility evaluation of items. As a result, the training time for hospital infection prevention and control was determined, two indicators were merged, six indicators were removed, six indicators were added, and one indicator was modified. In the second round, two indicators were added, one indicator was removed, and two indicators were merged. Following these consultations, the training model was finalized, consisting of four first-level indicators, 26 second-level indicators, and 44 third-level indicators. The first-level indicators encompassed the training content, training format and duration, assessment content, and evaluation indicators for hospital infection prevention and control. The weights of the indicators were calculated using the analytic hierarchy process. In descending order, the weights were assigned to the following: "training content, training format and

**Table 2. Results of expert suggestions' coordination degree (first round).**

| Hierarchical indicator | Kendall's W | $X^2$ | P |
|---|---|---|---|
| **Primary indicator** | 0.440 | 36.912 | <0.001 |
| **Secondary indicator** | 0.204 | 109.982 | <0.001 |
| **Tertiary indicator** | 0.386 | 237.402 | <0.001 |

**Table 3. Results of expert suggestions' coordination degree (second round).**

| Hierarchical indicator | Kendall's W | X² | P |
|---|---|---|---|
| Primary indicator | 0.562 | 33.703 | <0.001 |
| Secondary indicator | 0.467 | 119.023 | <0.001 |
| Tertiary indicator | 0.556 | 349.986 | <0.001 |

method, assessment content of the infectious disease knowledge system, and evaluation indicators for hospital infection prevention and control" (Table 4).

## Discussion

The standardized training system proposed for hospital infection prevention and control among newly recruited medical staff in the internal medicine ICU is both scientifically grounded and practical. It was developed based on the KAPB model, incorporating findings from literature research, interviews, the Delphi method, and other sources. The model consists of first-level indicators, including hospital infection prevention and control training content, training format and duration, assessment content, and evaluation indicators. It further encompasses 26 second-level indicators and 44 third-level indicators. Overall, this research system is scientifically robust and comprehensive, addressing the essential knowledge and skills required by new medical staff to effectively prevent and control nosocomial infections in the ICU. Among the 16 experts involved in the study, seven were renowned nationwide experts in hospital infection management, nine had expertise in intensive care. All of the selected experts possess extensive experience in ICU infection prevention and actively contribute to teaching and research in the field of ICU or nosocomial infection prevention and control. This underscores the reliability of the parameters chosen for the proposed model.

The index system developed in this study is comprised of four primary indicators, 26 secondary indicators, and 44 tertiary indicators. The weights of these indicators, listed in descending order, include: "training content, training form and method, assessment content of the infectious disease knowledge system, and evaluation indicators for hospital infection prevention and control." Of these, "Nosocomial infection prevention and control training content" bears the most significant weight, highlighting its importance in enabling new medical personnel to effectively prevent and control hospital infections in the ICU. This suggests a crucial need for mastery of the knowledge and skills related to hospital infection. In line with the theoretical framework of "knowing, trusting, and doing," proper knowledge training serves as the bedrock. Swift assimilation of theoretical knowledge bolsters infection prevention and control practices.

Within the secondary indicators, "patient management of multidrug-resistant bacteria" in "training content of hospital infection prevention and control" and "aseptic operation qualification rate" in "evaluation indicators" bore the highest weights. This suggests that managing multidrug resistance bacteria and implementing aseptic operations are vital components of infection prevention in the ICU, aligning with findings from other studies [23]. Common infections in the ICU, including ventilator-associated pneumonia, catheter-related bloodstream infections, and urinary tract infections [24], are intimately linked to invasive procedures. Consequently, this study incorporated pertinent infection prevention and control knowledge in theoretical training and emphasized aseptic techniques in skills operation training. This aims to fortify medical personnel's awareness of infection prevention and control and enhance their skills in aseptic technique to curb the spread of cross-infection in the ICU, ensure patient safety, and minimize ICU stay duration. Among the tertiary indicators, "hand

**Table 4. Construction of standardized training model for hospital infection prevention and control for new medical staff in the internal medicine ICU.**

| Index level | Significance grade | Variable coefficient | Weight targets | Combination weight targets |
|---|---|---|---|---|
| 1 Training content of hospital infection prevention and control | 4.93 ±0.25 | 0.05 | 0.450 | 0.450 |
| 1.1 Laws and Regulations on Nosocomial Infection | 4.56 ±0.63 | 0.14 | 0.036 | 0.016 |
| 1.1.1 "Law of the People's Republic of China on the Prevention and Treatment of Infectious Diseases" | 4.38 ±1.15 | 0.26 | 0.333 | 0.005 |
| 1.1.2 "Regulations on Emergency Response to Public Health Emergencies" | 4.56 ±0.89 | 0.2 | 0.667 | 0.011 |
| 1.2 Nosocomial infection monitoring | 4.56 ±0.63 | 0.14 | 0.037 | 0.017 |
| 1.2.1 Definition of nosocomial infection and reporting requirements | 4.81 ±0.40 | 0.08 | 0.500 | 0.008 |
| 1.2.2 Judgment of common nosocomial infections | 4.75 ±0.58 | 0.12 | 0.025 | 0.004 |
| 1.2.3 Nosocomial Infection Outbreak Definition and Reporting and Handling Process | 4.75 ±0.58 | 0.12 | 0.025 | 0.004 |
| 1.3 Disinfection and sterilization | 4.50 ±0.63 | 0.14 | 0.027 | 0.012 |
| 1.3.1 Basic principles of disinfection and sterilization | 4.69 ±0.70 | 0.15 | 0.667 | 0.008 |
| 1.3.2 Item classification and selection principles for disinfection and sterilization | 4.62 ±0.72 | 0.16 | 0.333 | 0.004 |
| 1.4 Hand hygiene | 4.81 ±0.40 | 0.08 | 0.075 | 0.034 |
| 1.4.1 Definition of hand hygiene | 4.81 ±0.54 | 0.11 | 0.198 | 0.007 |
| 1.4.2 Indications for hand hygiene | 4.88 ±0.34 | 0.07 | 0.312 | 0.011 |
| 1.4.3 Methods and precautions for hand hygiene | 4.94 ±0.25 | 0.05 | 0.490 | 0.017 |
| 1.5 Prevention and management of multidrug-resistant bacteria | 4.94 ±0.25 | 0.05 | 0.130 | 0.059 |
| 1.5.1 Definition of multi-drug resistant bacteria | 4.81 ±0.54 | 0.11 | 0.165 | 0.010 |
| 1.5.2 Key surveillance species of multidrug-resistant bacteria | 4.81 ±0.4 | 0.08 | 0.165 | 0.010 |
| 1.5.3 Judgment criteria for multidrug-resistant bacteria | 4.88 ±0.34 | 0.07 | 0.279 | 0.016 |
| 1.5.4 Prevention and control of multidrug-resistant bacteria | 4.94 ±0.25 | 0.05 | 0.392 | 0.023 |
| 1.6 Classification management of medical waste | 4.63 ±0.81 | 0.17 | 0.045 | 0.020 |
| 1.6.1 Definition and classification of medical waste | 4.63 ±0.89 | 0.19 | 0.667 | 0.013 |
| 1.6.2 Requirements for Medical Waste Management | 4.56 ±0.89 | 0.2 | 0.333 | 0.007 |
| 1.7 Occupational protection | 4.88 ±0.34 | 0.07 | 0.107 | 0.048 |
| 1.7.1 Definition of Standard Precautions | 4.81 ±0.4 | 0.08 | 0.185 | 0.009 |
| 1.7.2 Selection and use of protective equipment | 4.94 ±0.25 | 0.05 | 0.433 | 0.021 |
| 1.7.3 Handling of occupational exposures | 4.88 ±0.52 | 0.1 | 0.321 | 0.015 |
| 1.7.4 Safe injection | 4.81 ±0.41 | 0.08 | 0.185 | 0.009 |
| 1.8 Ventilator-associated pneumonia prevention and control | 4.81 ±0.41 | 0.08 | 0.289 | 0.076 |
| 1.8.1 Definition of Ventilator-Associated Pneumonia | 4.63 ±0.62 | 0.13 | 0.297 | 0.018 |
| 1.8.2 Diagnosis of ventilator-associated pneumonia | 4.5 ±0.63 | 0.14 | 0.164 | 0.010 |
| 1.8.3 Prevention and Control Strategies of Ventilator-Associated Pneumonia | 4.81 ±0.41 | 0.08 | 0.539 | 0.033 |
| 1.9 Prevention and control of catheter-related bloodstream infection | 4.4 ±0.72 | 0.16 | 0.057 | 0.026 |
| 1.9.1 Definition of catheter-related bloodstream infection | 4.75 ±0.45 | 0.09 | 0.312 | 0.008 |
| 1.9.2 Diagnosis of catheter-related bloodstream infection | 4.56 ±0.63 | 0.14 | 0.198 | 0.005 |
| 1.9.3 Prevention and control strategies for catheter-related bloodstream infections | 4.81 ±0.4 | 0.08 | 0.491 | 0.013 |
| 1.10 Prevention and control of catheter-related urinary tract infections | 4.44 ±0.73 | 0.16 | 0.051 | 0.023 |
| 1.10.1 Definition of catheter-related urinary tract infection | 4.63 ±0.62 | 0.13 | 0.312 | 0.007 |
| 1.10.2 Diagnosis of catheter-related urinary tract infection | 4.56 ±0.51 | 0.11 | 0.198 | 0.004 |
| 1.10.3 Prevention and control strategies for catheter-related urinary tract infections | 4.75 ±0.44 | 0.09 | 0.491 | 0.011 |
| 1.11 Infectious disease knowledge | 4.5±0.18 | 0.04 | 0.260 | 0.260 |
| 1.12 Aseptic technique (Operation) | 4.81±0.85 | 0.17 | 0.154 | 0.017 |
| 1.13 Cleaning and Disinfection (Operation) | 4.50 ±0.10 | 0.02 | 0.075 | 0.034 |
| 1.13.1 Disinfection of instruments and equipment | 4.69 ±0.60 | 0.13 | 0.500 | 0.017 |
| 1.13.2 Terminal disinfection of bed units | 4.69 ±0.60 | 0.13 | 0.500 | 0.017 |
| 1.14 Standard Prevention Related Skills (Operations) | 4.88 ±0.09 | 0.02 | 0.109 | 0.049 |

(*Continued*)

**Table 4.** (Continued)

| Index level | Significance grade | Variable coefficient | Weight targets | Combination weight targets |
|---|---|---|---|---|
| 1.14.1 hand hygiene | 4.94 ±0.25 | 0.05 | 0.500 | 0.024 |
| 1.14.2 Correct putting on and taking off of gloves, masks, isolation gowns, protective clothing and other protective equipment | 4.94 ±0.25 | 0.05 | 0.500 | 0.024 |
| 1.15 Specimen Collection (Operation) | 4.75 ±0.11 | 0.02 | 0.057 | 0.026 |
| 1.15.1 Collection of blood samples | 4.75 ±0.45 | 0.09 | 0.381 | 0.007 |
| 1.15.2 Sputum specimen collection | 4.75 ±0.45 | 0.09 | 0.381 | 0.007 |
| 1.15.3 Urine specimen collection | 4.69 ±0.6 | 0.13 | 0.381 | 0.007 |
| 1.15.4 Oropharyngeal/nasopharyngeal swab collection | 4.75 ±0.45 | 0.09 | 0.191 | 0.004 |
| 2. Training methods/formats | 4.50 ±0.73 | 0.16 | 0.260 | 0.260 |
| 2.1 Theory lecture + operation demonstration or video | 4.75 ±0.11 | 0.02 | 0.667 | 0.173 |
| 2.2 Clinical case discussions or scenario presentations | 4.69 ±0.15 | 0.03 | 0.333 | 0.087 |
| 3. Evaluation content | 4.50 ±0.73 | 0.16 | 0.171 | 0.171 |
| 3.1 Theoretical knowledge examination | 4.44 ±0.73 | 0.16 | 0.333 | 0.057 |
| 3.2 Operational skills examination | 4.63 ±0.62 | 0.13 | 0.667 | 0.114 |
| 3.2.1 Hand hygiene | 4.94 ±0.25 | 0.05 | 0.215 | 0.025 |
| 3.2.2 Putting on and taking off masks and gloves | 4.94 ±0.25 | 0.05 | 0.215 | 0.025 |
| 3.2.3 Putting on and taking off the gown | 4.81 ±0.41 | 0.08 | 0.154 | 0.017 |
| 3.2.4 Collection of blood samples | 4.81 ±0.41 | 0.08 | 0.154 | 0.017 |
| 3.2.5 Sputum specimen collection | 4.69 ±0.48 | 0.1 | 0.097 | 0.011 |
| 3.2.6 Urine specimen collection | 4.69 ±0.48 | 0.1 | 0.097 | 0.011 |
| 3.2.7 Oropharyngeal/nasopharyngeal swab collection | 4.44 ±0.73 | 0.16 | 0.068 | 0.008 |
| 4. Evaluation Indicators | 4.43 ±0.64 | 0.14 | 0.120 | 0.120 |
| 4.1Qualification rate of knowledge of hospital infection prevention and control | 4.64 ±0.62 | 0.13 | 0.114 | 0.014 |
| 4.2Hand hygiene compliance and correctness rate | 4.88 ±0.34 | 0.07 | 0.203 | 0.024 |
| 4.3Qualified rate of aseptic operation | 4.94 ±0.25 | 0.05 | 0.265 | 0.032 |
| 4.4Correct implementation rate of occupational protection | 4.75 ±0.45 | 0.09 | 0.157 | 0.019 |
| 4.5Number of non-compliant operations | 4.44 ±0.73 | 0.16 | 0.072 | 0.009 |
| 4.6 Incidence of occupational exposure | 4.50 ±0.82 | 0.18 | 0.088 | 0.011 |
| 4.7 Course Training Satisfaction | 4.31 ±0.79 | 0.18 | 0.044 | 0.005 |

hygiene," "prevention and control of multidrug-resistant bacteria," and "correct donning and doffing of protective equipment" were given the highest weight. Firstly, physical contacts and hand hygiene play pivotal roles in indirect transmission, as most hospital-acquired infections spread via this mode. Despite hand hygiene being deemed as the most fundamental, effective, and important measure for nosocomial infection prevention and control, medical staff compliance remains suboptimal. This underscores the significance of reinforcing hand hygiene training. Secondly, multidrug-resistant bacterial infection poses a global threat to patient safety and healthcare quality, and targeted training for new medical staff can mitigate problems associated with multidrug-resistant bacteria [25, 26]. Lastly, correctly wearing and removing protective equipment is an essential step in infection prevention [27] and a vital skill for emergency rescue [28]. In summary, it is advised to prioritize protective skills during training for new medical personnel to effectively disrupt the transmission pathways of infectious diseases and reduce occupational exposure incidents [29].

The shortage of clinical medical staff, coupled with high workloads and strenuous tasks, impedes the swift completion of high-quality hospital infection prevention and control training. Taking into account the constrained training timeframe and the practicality of

implementation, in conjunction with the basic requisites for hospital infection prevention and control in the ICU, this study revisited and refined the training curriculum. This was achieved through two rounds of expert consultation, with an augmented focus on the core knowledge and pivotal skills required for hospital infection prevention and control. This enhancement aids medical personnel in rapidly acquiring key knowledge and skills, thus, reducing learning duration. It facilitates the swift integration of knowledge into practice, thereby advancing the awareness and capacity of new ICU staff in the prevention and control of nosocomial infections.

Nonetheless, this study is not without limitations. The consulted experts hail solely from two provinces in China, and the initially constructed training system is still predominantly within the theoretical framework. Future steps involve undertaking clinical applications and broadening the research scope to verify its scientific validity and efficacy, particularly in medical settings experiencing a scarcity of medical personnel.

## Conclusion

Drawing upon the KAPB model, this study engineered a standardized training system for hospital infection prevention and control. This system targets new medical personnel within the internal medicine ICU and was developed via literature research, qualitative interviews, expert consultations, and the AHP. The method of construction is rigorous, and the training content is both targeted and comprehensive, encompassing various facets of ICU hospital infection management. This includes pertinent theoretical knowledge and practical operational training aimed at enhancing the overall competency of new ICU medical staff in preventing and controlling hospital infections. In addition, it can strengthen patient safety and reduce healthcare-associated infections.

## Supporting information

**S1 Data.**
(ZIP)

## Acknowledgments

Thank you to all the experts who participated in this study during their busy schedules.

## Author Contributions

**Conceptualization:** Li Tang.

**Data curation:** Linfei Wu, Min Liu.

**Formal analysis:** Linli Zhuang.

**Investigation:** Linfei Wu, Min Liu, Jianfang Li.

**Methodology:** Linfei Wu, Linli Zhuang.

**Resources:** Li Tang.

**Software:** Wenyi Xie.

**Supervision:** Li Tang.

**Writing – original draft:** Linfei Wu.

**Writing – review & editing:** Wenyi Xie.

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
