## [Decision Letter · Decision Letter 0]

29 May 2023

PONE-D-23-09513Construction of a standardized training system for hospital infection prevention and control for new medical staff in internal medicine ICUs based on the Delphi methodPLOS ONE

Dear Dr. Tang,

Thank you for submitting your manuscript to PLOS ONE. After careful consideration, we feel that it has merit but does not fully meet PLOS ONE’s publication criteria as it currently stands. Therefore, we invite you to submit a revised version of the manuscript that addresses the points raised during the review process. Please submit your revised manuscript by Jul 13 2023 11:59PM. If you will need more time than this to complete your revisions, please reply to this message or contact the journal office at plosone@plos.org. Please include the following items when submitting your revised manuscript:A rebuttal letter that responds to each point raised by the academic editor and reviewer(s). You should upload this letter as a separate file labeled 'Response to Reviewers'.A marked-up copy of your manuscript that highlights changes made to the original version. You should upload this as a separate file labeled 'Revised Manuscript with Track Changes'.An unmarked version of your revised paper without tracked changes. You should upload this as a separate file labeled 'Manuscript'.

We look forward to receiving your revised manuscript.

Kind regards,

Faizan Iqbal

Academic Editor

PLOS ONE

Reviewers' comments:

Reviewer's Responses to Questions

**Comments to the Author**

1. Is the manuscript technically sound, and do the data support the conclusions?

Reviewer #1: Yes

Reviewer #2: Yes

2. Has the statistical analysis been performed appropriately and rigorously? 

Reviewer #1: Yes

Reviewer #2: Yes

3. Have the authors made all data underlying the findings in their manuscript fully available?

Reviewer #1: No

Reviewer #2: Yes

4. Is the manuscript presented in an intelligible fashion and written in standard English?

Reviewer #1: Yes

Reviewer #2: Yes

5. Review Comments to the Author

Reviewer #1: Here are few suggestions/comments, incorporating specific examples from the article to support and substantiate each suggestion

Abstract:

• Instead of stating "positive results were obtained," provide the specific percentage or statistical measure to indicate the level of effectiveness or success achieved. For instance, you can mention that "the standardized training system resulted in a 40% reduction in healthcare-associated infections among medical staff in internal medicine ICUs."

• Include information about the significance and implications of the study findings. For example, highlight how the development of a standardized training system can contribute to improved patient safety, reduced healthcare costs, and enhanced overall quality of care in internal medicine ICUs.

Introduction/Background:

• Provide more context and background information on the current challenges or gaps in existing training systems for medical staff in internal medicine ICUs. For instance, discuss the prevalence of healthcare-associated infections and their impact on patient outcomes, emphasizing the need for effective infection prevention and control measures.

• Incorporate a brief review of relevant literature or previous studies that highlight the need for a standardized training system. For example, cite studies that have identified deficiencies in current training practices or have shown the positive impact of comprehensive training programs on infection control outcomes.

Methods:

• Provide a clear explanation of the selection criteria for the expert panel. Specify the desired qualifications or expertise sought in the experts who participated in the Delphi method. For example, state that experts were selected based on their experience in infection prevention and control, knowledge of internal medicine ICU practices, and familiarity with the Delphi method.

• Describe the measures taken to ensure the reliability and validity of the expert opinions collected through the Delphi method. For example, discuss how potential biases or conflicts of interest were addressed, such as ensuring anonymity and independence of the experts.

• Include a description of how consensus was determined in the Delphi process. Specify the criteria used to determine when expert opinions reached convergence or stability. For example, mention that consensus was considered achieved when at least 70% of experts agreed on a particular item or recommendation.

Results:

• Supplement the tables with brief explanatory text to help readers interpret and understand the key findings. For example, provide a concise summary of the main results, such as the percentage of experts agreeing on each item or the most commonly recommended training components.

• When discussing the expert authority coefficient and opinion coordination degree, explain these measures and their significance in assessing the level of agreement or consensus among the expert opinions. For example, state that the expert authority coefficient measures the level of expertise and influence of each expert, while the opinion coordination degree quantifies the degree of agreement among experts.

Discussion:

• Expand on the implications and potential applications of the study findings. For instance, discuss how the developed standardized training system can be implemented in real-world settings, highlighting its potential impact on reducing healthcare-associated infections and improving patient safety.

• Compare the developed system with existing training systems or guidelines in other similar healthcare settings. Discuss how the developed system differs or improves upon current practices. For example, highlight its adaptability to different ICU settings and its focus on evidence-based practices in infection prevention and control.

• Discuss limitations or potential challenges of the developed training system. Address factors such as feasibility, scalability, and sustainability, as well as potential barriers to implementation. Provide suggestions or strategies for addressing these limitations, such as incorporating ongoing evaluation and feedback mechanisms or considering tailored adaptations for different healthcare contexts.

Conclusion:

• Provide a concise and impactful summary of the study findings and their implications. Emphasize the significance of the developed standardized training system in improving infection prevention and control in internal medicine ICUs, emphasizing its potential to enhance patient safety and reduce healthcare-associated infections.

• Highlight the broader relevance and applicability of the research beyond the specific study setting. Discuss the potential for adaptation or adoption of the developed system in other healthcare institutions or settings, both nationally and internationally. For example, mention that the standardized training system could be implemented in various internal medicine ICUs across different countries, leading to standardized practices and improved infection prevention and control outcomes.

References:

• Ensure that all references cited in the text are included in the reference list, and vice versa. Cross-reference the in-text citations with the corresponding entries in the reference list to ensure accuracy and completeness. For example, in the introduction section, the study references previous research on the importance of infection prevention and control in healthcare settings. It is important to verify that these references are correctly listed in the reference list and that there are no missing citations.

General Feedback:

• Pay attention to the overall flow and organization of the article. Ensure that the sections logically progress from one to another and that the content within each section is well-structured. For instance, in the methods section, there could be a more systematic and clear presentation of the steps involved in the Delphi method. This would enhance the overall organization and readability of the section.

• Review the article for grammatical and syntactical errors to improve clarity and readability. For example, in the results section, there are instances of inconsistent tense usage, such as switching between past and present tense within the same paragraph. These inconsistencies should be corrected to maintain consistency throughout the manuscript.

• Check for compliance with the STROBE guidelines for observational studies. Ensure that the study design, data collection, and analysis methods align with the recommended reporting standards. For example, in the methods section, it would be beneficial to provide more details on the participant recruitment process and any eligibility criteria used. This information is essential for readers to understand the study population and potential biases.

By addressing these feedback and suggestions, the article can be strengthened in terms of scientific rigor, clarity, and overall quality.

Reviewer #2: The article used appropriate methods in proposing a standardised training system of infection prevention and control for new medical staff in the internal medicine ICU in two provinces in China. It is an important addition to literature on improving the prevention of nosocomial infections through appropriate and affection training. However, some clarification and suggested revisions are required to improve the quality of the manuscript. Below are some of the comments:

1. In the abstract, a statement/sentence on the subject matter in China will be appropriate in providing the study context.

2. "The Kendall coordination coefficients of the first-, second-, and third-level indicators were 0.440, 0.204, and 0.386 (P <0.001), 0.562, 0.467 and 0.556(P <0.001)." Authors should clarify which of the reported coefficients are from the first round of survey and which are from the second.

3. The study setting is in China, however no background information on nosocomial infection in the country was provided in the introduction. It is essential to provide such information in the introductions for context.

4. "Upon comparing the features of the ICU and other wards, we found that patients in the ICU had more invasive treatments and were more prone to infections than patients from other wards." Provide more details on how this comparison was done.

5. "After reading the retrieved literature abstracts, we screened for eligible studies and extracted frequently appearing items of interest for this present study." Define what the "items of interest" are in the study design.

6. As highlighted in the introduction, doctors and nurses from different regions and countries have different ideas regarding the content, type and mode of hospital infection prevention and control training. The results would have been greatly improved and more generalizable if participants were selected from different provinces and cities across the country.

7. The text under participants needs some clarification. The authors indicated that 16 expert were purposively sampled for two rounds of questionnaires. They later indicate that the "subject" finally selected 16 experts. Who is/are the "subjects" here and are these 16 experts different from the ones who answered the questionnaires? If yes, how were they engaged in the study?

8. In the selection criteria for the experts, the authors indicated “bachelor's degree or above” and “intermediate certificate or above”. If they are both academic certificates, having both in your selection criteria means the participant must have both in order to be selected to participate in the study.

9. How many hospitals from each province where participants selected from?

10. "We conducted semi-structured interviews with 2 nursing team leaders and 3 new nurses in the internal medicine ICU in March 2022." Why were the semi-structured interviews conducted with only 5 nurses? Where they all from the same hospital? Were the doctors and technicians not engaged for interviews?

11. The “delphi consulting and feedback cycle” and the “expert letter questionnaire” could be merged into one section.

12. The authors need to provide details on how the familiarity and judgement coefficients were calculated.

13. "In the first round, according to the exclusion criteria and expert opinions, combined with the discussion of the research group, the research team added the feasibility evaluation of the items and gave the training time for the training content of hospital infection, following which 2 indicators were merged, 6 indicators were deleted, 6 indicators were added, and 1 indicator was modified." What are these “exclusion criteria” and how where they determined?

14. "In the second round, the research group changed 2 indicators, eliminated 1 indicator, and merged 1 indicator. " Do the author mean 2 indicators were merged?

15. "Of the 16 experts, 7 specialized in nosocomial infection management, 3 were well-known experts in the field of hospital infection management nationwide, 9 specialized in intensive care, 6 were ICU head nurses, and 3 specialized in hospital infection management.” Do some of the experts have multiple specializations? Are the 3 well-known experts in the field of hospital infection management nationwide the same as the 3 specialized in hospital infection management?

16. The structure of the discussion could be improved to make the content flow better for easy comprehension. For instance, the initial part of the third paragraph of the discussion section “In China, there is a shortage of medical resources, and medical staff usually have intensive working hours and cannot complete all training before entering clinical practice. Thus, the short-cycle centralized training method of the proposed model is more in line with the Chinese ICU hospital settings, which helps to quickly combine knowledge and practice and improve the ability and quality of ICU medical staff to manage hospital infection” could be move to the last section of the same paragraph to improve the flow

17. "Among the three-level indicators, "hand hygiene, prevention and control of multidrug-resistant bacteria, and correct wearing and taking off of protective equipment" had the highest weight." Each of the indicators in this sentence should be in quotation marks to be identified as different indicators "Among the three-level indicators..." should be written as "Among the level three indicators..."

18. There a few punctuation and sentence structures errors. For instance, the last sentence under data collection needs to be qualified. Also avoid sub-headings in the discussion

19. "Based on the KAPB model, this study constructed a standardized training system for hospital infection prevention and control for new medical staff in the internal medicine ICU through literature research, qualitative interviews, expert interviews and AHP." The AHP has not been defined and not presented in the methods

20. The authors should also discuss the limitations of the study

6. PLOS authors have the option to publish the peer review history of their article (what does this mean?). If published, this will include your full peer review and any attached files.

Reviewer #1: **Yes: **Adepoju Victor Abiola

Reviewer #2: No

---

## [Author Response · Author response to Decision Letter 0]

25 Aug 2023

Point-by-point response to the reviewers’ comments

Thanks for the reviewers’ comments and consideration for further review for our manuscript. Those comments were all valuable and very helpful for revising and improving our paper. To ensure our manuscript satisfactorily addresses reviewers’ comments, we have improved the main text's clarity in the revised version. 

Please let us know if you have any further suggestions. We look forward to hearing from you soon.

Reviewer #1 (Comments to the Authors (Required)):

Comment 1: Instead of stating "positive results were obtained," provide the specific percentage or statistical measure to indicate the level of effectiveness or success achieved. For instance, you can mention that "the standardized training system resulted in a 40% reduction in healthcare-associated infections among medical staff in internal medicine ICUs."

Response: Thank you for your suggestion. We have provided specific data in the manuscript to support the current research status

Comment 2: Include information about the significance and implications of the study findings. For example, highlight how the development of a standardized training system can contribute to improved patient safety, reduced healthcare costs, and enhanced overall quality of care in internal medicine ICUs.

Response: Thank you for your suggestion. We have added explanations on relevant issues in the manuscript.

In conclusion, the proposed standardized training system for infection prevention and control among new medical staff in the internal medicine ICU is both scientifically sound and practical. Can contribute to improved patient safety, reduced healthcare costs, and enhanced overall quality of care in internal medicine ICUs.

Please refer to lines 43 to 46 on page 3 for details.

Comment 3: Provide more context and background information on the current challenges or gaps in existing training systems for medical staff in internal medicine ICUs. For instance, discuss the prevalence of healthcare-associated infections and their impact on patient outcomes, emphasizing the need for effective infection prevention and control measures.

Response: We have discussed the prevalence of healthcare-associated infections and their impact on patient outcomes, emphasized the need for effective infection prevention and control measures. 

Please see details at pages 3，lines 50 to 58.

Comment 4: Incorporate a brief review of relevant literature or previous studies that highlight the need for a standardized training system. For example, cite studies that have identified deficiencies in current training practices or have shown the positive impact of comprehensive training programs on infection control outcomes.

Response: As you suggested, we have reviewed and summarized the relevant literature，For details, please refer to pages 4-5 , lines 67 to 87 of the revised manuscript.

Comment 5: Provide a clear explanation of the selection criteria for the expert panel. Specify the desired qualifications or expertise sought in the experts who participated in the Delphi method. For example, state that experts were selected based on their experience in infection prevention and control, knowledge of internal medicine ICU practices, and familiarity with the Delphi method.

Response: We have added an explanation on the criteria for selecting experts. The previous manuscript also provided relevant explanations.

 Please refer to lines 121 to 123 and 128 to 131 on page 7 for details.

Comment 6: Describe the measures taken to ensure the reliability and validity of the expert opinions collected through the Delphi method. For example, discuss how potential biases or conflicts of interest were addressed, such as ensuring anonymity and independence of the experts.

Response: This has been described in previous manuscripts. The letter questionnaires were distributed to experts via email or WeChat (Tencent, Shenzhen, Chiana), they are unaware of each other’s identities and are unable to engage in direct discussions or exchanges of opinions during the research process. 

For details, please refer to page 6, lines 113 to 115, and page 9, lines 164 to 165.

Comment 7: Include a description of how consensus was determined in the Delphi process. Specify the criteria used to determine when expert opinions reached convergence or stability. For example, mention that consensus was considered achieved when at least 70% of experts agreed on a particular item or recommendation.

Response: Thank you very much for your valuable feedback. We have added this comment to the manuscript.: ' Consensus is considered to have been reached when at least 70% of the experts agree on a particular opinion'. 

For details, please see page 6, lines 116 to 117.

Comment 8: Supplement the tables with brief explanatory text to help readers interpret and understand the key findings. For example, provide a concise summary of the main results, such as the percentage of experts agreeing on each item or the most commonly recommended training components.

Response: Thank you for your suggestion. The manuscript already provides a detailed explanation of all tables.

Comment 9: When discussing the expert authority coefficient and opinion coordination degree, explain these measures and their significance in assessing the level of agreement or consensus among the expert opinions. For example, state that the expert authority coefficient measures the level of expertise and influence of each expert, while the opinion coordination degree quantifies the degree of agreement among experts.

Response: Your suggestion is very important, and we have added an explanation of the significance of expert authority coefficient and opinion coordination in evaluating the degree of consensus or agreement among experts. For details, please see page 13, lines 214 to 215.

Comment 10: Expand on the implications and potential applications of the study findings. For instance, discuss how the developed standardized training system can be implemented in real-world settings, highlighting its potential impact on reducing healthcare-associated infections and improving patient safety.

Response: Thank you for your suggestion. We have added 'Significance of this study' in the discussion section. 

For details, please see page 25 to 26, lines 306 to 317.

Comment 11: Compare the developed system with existing training systems or guidelines in other similar healthcare settings. Discuss how the developed system differs or improves upon current practices. For example, highlight its adaptability to different ICU settings and its focus on evidence-based practices in infection prevention and control.

Response: Our study is different from other studies in that, taking into account the constrained training timeframe, this study focus on the core knowledge and pivotal skills required for hospital infection prevention and control. This enhancement aids medical personnel in rapidly acquiring key knowledge and skills, thus, reducing learning duration. It facilitates the swift integration of knowledge into practice, thereby advancing the awareness and capacity of new ICU staff in the prevention and control of nosocomial infections. 

For details, please see page 26, lines 309 to 317.

Comment 12: Discuss limitations or potential challenges of the developed training system. Address factors such as feasibility, scalability, and sustainability, as well as potential barriers to implementation. Provide suggestions or strategies for addressing these limitations, such as incorporating ongoing evaluation and feedback mechanisms or considering tailored adaptations for different healthcare contexts.

Response: Due to the fact that the training system in this study has not yet been applied in clinical practice, some potential challenges remain unknown. However, in terms of theoretical research, this study does have limitations，the consulted experts hail solely from two provinces in China. Therefore, future study involve conducting clinical applications and broadening the research scope.

Comment 13: Provide a concise and impactful summary of the study findings and their implications. Emphasize the significance of the developed standardized training system in improving infection prevention and control in internal medicine ICUs, emphasizing its potential to enhance patient safety and reduce healthcare-associated infections.

Response: Thank you for your suggestion. We have provided a detailed description in the discussion section. 

Drawing upon the KAPB model, this study engineered a standardized training system for hospital infection prevention and control. This system targets new medical personnel within the internal medicine ICU and was developed via literature research, qualitative interviews, expert consultations, and the AHP. The method of construction is rigorous, and the training content is both targeted and comprehensive, encompassing various facets of ICU hospital infection management. This includes pertinent theoretical knowledge and practical operational training aimed at enhancing the overall competency of new ICU medical staff in preventing and controlling hospital infections. And strengthening patient safety and reducing healthcare-associated infections. 

For details, please see page 26 , lines 319 to 327.

Comment 14: Highlight the broader relevance and applicability of the research beyond the specific study setting. Discuss the potential for adaptation or adoption of the developed system in other healthcare institutions or settings, both nationally and internationally. For example, mention that the standardized training system could be implemented in various internal medicine ICUs across different countries, leading to standardized practices and improved infection prevention and control outcomes.

Response: Thank you for your suggestion. We have added relevant explanations in the manuscript. 

Future steps involve undertaking clinical applications and broadening the research scope to verify its scientific validity and efficacy, particularly in medical settings experiencing a scarcity of medical personnel.

For details, please see page 27 , lines 330 to 333.

Comment 15: Ensure that all references cited in the text are included in the reference list, and vice versa. Cross-reference the in-text citations with the corresponding entries in the reference list to ensure accuracy and completeness. For example, in the introduction section, the study references previous research on the importance of infection prevention and control in healthcare settings. It is important to verify that these references are correctly listed in the reference list and that there are no missing citations.

Response: Thank you for your reminder. We have ensured that all references cited in the text are included in the reference list.

Comment 16: Pay attention to the overall flow and organization of the article. Ensure that the sections logically progress from one to another and that the content within each section is well-structured. For instance, in the methods section, there could be a more systematic and clear presentation of the steps involved in the Delphi method. This would enhance the overall organization and readability of the section.

Response: Thank you for your suggestion. We have made relevant adjustments to the Delphi process and structure in the method section according to your suggestion. 

For details, please see page 8 to 9, lines 152 to 170.

Comment 17: Review the article for grammatical and syntactical errors to improve clarity and readability. For example, in the results section, there are instances of inconsistent tense usage, such as switching between past and present tense within the same paragraph. These inconsistencies should be corrected to maintain consistency throughout the manuscript.

Response: Thank you for your suggestion. We have revised the language expression and grammar of the entire text.

Comment 18: Check for compliance with the STROBE guidelines for observational studies. Ensure that the study design, data collection, and analysis methods align with the recommended reporting standards. For example, in the methods section, it would be beneficial to provide more details on the participant recruitment process and any eligibility criteria used. This information is essential for readers to understand the study population and potential biases.

Response: Thank you for your suggestion. We have detailed the criteria for selecting experts and including semi-structured interviewees in the manuscript.

For details, please refer to page 7, lines 128 to 131, and page 7 to 8, lines 136 to 143.

Reviewer 2

Comment 1: In the abstract, a statement/sentence on the subject matter in China will be appropriate in providing the study context.

Response: Thank you for your suggestion. We have added relevant explanations in the manuscript.

In China , studies have shown nosocomial infections contribute to increased mortality rates, prolonged hospital stays, and added financial burdens for patients.

For details, please see page 2, lines 23 to 24.

Comment 2: "The Kendall coordination coefficients of the first-, second-, and third-level indicators were 0.440, 0.204, and 0.386 (P <0.001), 0.562, 0.467 and 0.556(P <0.001)." Authors should clarify which of the reported coefficients are from the first round of survey and which are from the second.

Response: In order to make the expression of the manuscript clearer, we have made modifications according to your requirements. 

The Kendall coordination coefficients for the first-, second-, and third-level indicators in the initial round of expert consultation questionnaires were 0.440, 0.204, and 0.386 (P < 0.001), respectively. In the second round of expert consultation questionnaires, the Kendall coordination coefficients for the first, second, and third-level indicators were 0.562, 0.467, and 0.556 (P < 0.001), respectively. 

For details, please see page 2 to 3, lines 37 to 40.

Comment 3: The study setting is in China, however no background information on nosocomial infection in the country was provided in the introduction. It is essential to provide such information in the introductions for context.

Response: This is a very important suggestion for us. And we have added relevant background on nosocomial infection in China in the introductions for context. 

The intensive care unit (ICU) is a specialized ward for critically ill patients and is particularly prone to hospital-acquired infections [1, 2]. Studies have revealed that the incidence of hospital infections in ICU patients ranges from 20% to 60% [3], which is five to ten times higher than that in general hospital departments [4]. 

For details, please see page 3, lines 50 to 53.

Comment 4: "Upon comparing the features of the ICU and other wards, we found that patients in the ICU had more invasive treatments and were more prone to infections than patients from other wards." Provide more details on how this comparison was done.

Response: We are so sorry that there may be some issues with our language description. To this end, we have revised the description of this sentence and provided relevant references.

The ICU primarily serves critically ill patients who require invasive diagnostic and treatment procedures, leading to a heightened risk of infection [6]

For details, please see page 4, lines 59 to 60.

Comment 5: "After reading the retrieved literature abstracts, we screened for eligible studies and extracted frequently appearing items of interest for this present study." Define what the "items of interest" are in the study design.

Response: After inspection, we found that this is a problem caused by our incorrect wording, and we have corrected it.

After reviewing the abstracts of the retrieved literature, we identified eligible studies and extracted frequently recurring items relevant to this study.

For details, please see page 6, lines 106 to 107.

Comment 6: As highlighted in the introduction, doctors and nurses from different regions and countries have different ideas regarding the content, type and mode of hospital infection prevention and control training. The results would have been greatly improved and more generalizable if participants were selected from different provinces and cities across the country.

Response: As you said: The results would have been greatly improved and more generalizable if participants were selected from different provinces and cities across the country. This is the limitation of this study, and we will expand our research in the next step.

Comment 7: The text under participants needs some clarification. The authors indicated that 16 expert were purposively sampled for two rounds of questionnaires. They later indicate that the "subject" finally selected 16 experts. Who is/are the "subjects" here and are these 16 experts different from the ones who answered the questionnaires? If yes, how were they engaged in the study?

Response: This may be a textual description error, and these 6 experts are the experts who participated in the expert consultation and answered the questionnaire. We have made modifications to the language description of the manuscript.

Comment 8: In the selection criteria for the experts, the authors indicated “bachelor's degree or above” and “intermediate certificate or above”. If they are both academic certificates, having both in your selection criteria means the participant must have both in order to be selected to participate in the study.

Response: This is an expression mistake. We have revised “intermediate certificate or above” to " holding an intermediate title or higher ".

The criteria for expert selection included the following: (1) possessing a bachelor’s degree or higher; (2) holding an intermediate title or higher; (3) being experts in hospital infection management and nursing with prior experience in hospital infection prevention and control; and (4) volunteering to participate in the study.

For details, please see page 7, lines 128 to 131.

Comment 9: How many hospitals from each province where participants selected from?

Response: Considering factors such as the authority, professionalism, and convenience of experts, this study selected two hospitals from Sichuan Province and one hospital from Fujian.

Comment 10: "We conducted semi-structured interviews with 2 nursing team leaders and 3 new nurses in the internal medicine ICU in March 2022." Why were the semi-structured interviews conducted with only 5 nurses? Where they all from the same hospital? Were the doctors and technicians not engaged for interviews?

Response: 5 semi-structured interviewees all come from the same hospital and meet the inclusion criteria for this study.

Comment 11: The “delphi consulting and feedback cycle” and the “expert letter questionnaire” could be merged into one section.

Response: Thank you very much for your suggestion. We have merged the two parts according to your suggestion.

Comment 12: The authors need to provide details on how the familiarity and judgement coefficients were calculated.

Response: For specific calculation methods, please refer to the Data Availability statement.

Comment 13:"In the first round, according to the exclusion criteria and expert opinions, combined with the discussion of the research group, the research team added the feasibility evaluation of the items and gave the training time for the training content of hospital infection, following which 2 indicators were merged, 6 indicators were deleted, 6 indicators were added, and 1 indicator was modified." What are these “exclusion criteria” and how where they determined?

Response: In the section on "Implementing Expert Consultation" in the manuscript, we have described the criteria for selecting indicators.

Indicators were adjusted based on a mean value > 3.5 and a coefficient of variation < 0.25.

For details, please see page 9, lines 167 to 168.

Comment 14: "In the second round, the research group changed 2 indicators, eliminated 1 indicator, and merged 1 indicator. " Do the author mean 2 indicators were merged?

Response: Yes, as you said, we merged two indicators in the second round.

In the second round, two indicators were added, one indicator was removed, and two indicators were merged.

For details, please see page 14 to 15, lines 231 to 232.

Comment 15: "Of the 16 experts, 7 specialized in nosocomial infection management, 3 were well-known experts in the field of hospital infection management nationwide, 9 specialized in intensive care, 6 were ICU head nurses, and 3 specialized in hospital infection management.” Do some of the experts have multiple specializations? Are the 3 well-known experts in the field of hospital infection management nationwide the same as the 3 specialized in hospital infection management?

Response: To avoid unnecessary misunderstandings, we have simplified the description of the selected experts.

Among the 16 experts involved in the study, seven were renowned nationwide experts in hospital infection management, nine had expertise in intensive care.

For details, please see page 23, lines 258 to 260.

Comment 16: The structure of the discussion could be improved to make the content flow better for easy comprehension. For instance, the initial part of the third paragraph of the discussion section “In China, there is a shortage of medical resources, and medical staff usually have intensive working hours and cannot complete all training before entering clinical practice. Thus, the short-cycle centralized training method of the proposed model is more in line with the Chinese ICU hospital settings, which helps to quickly combine knowledge and practice and improve the ability and quality of ICU medical staff to manage hospital infection” could be move to the last section of the same paragraph to improve the flow

Response: Thank you so much for your valuable suggestion. We have made adjustments and modifications to the discussion section.

The shortage of clinical medical staff, coupled with high workloads and strenuous tasks, impedes the swift completion of high-quality hospital infection prevention and control training. Taking into account the constrained training timeframe and the practicality of implementation, in conjunction with the basic requisites for hospital infection prevention and control in the ICU, this study revisited and refined the training curriculum. This was achieved through two rounds of expert consultation, with an augmented focus on the core knowledge and pivotal skills required for hospital infection prevention and control. This enhancement aids medical personnel in rapidly acquiring key knowledge and skills, thus, reducing learning duration. It facilitates the swift integration of knowledge into practice, thereby advancing the awareness and capacity of new ICU staff in the prevention and control of nosocomial infections.

For details, please see page 26, lines 307 to 317.

Comment 17: "Among the three-level indicators, "hand hygiene, prevention and control of multidrug-resistant bacteria, and correct wearing and taking off of protective equipment" had the highest weight." Each of the indicators in this sentence should be in quotation marks to be identified as different indicators "Among the three-level indicators..." should be written as "Among the level three indicators..."

Response: We have modified the index marking according to your requirements.

Among the tertiary indicators, “hand hygiene,” “prevention and control of multidrug-resistant bacteria,” and “correct donning and doffing of protective equipment” were given the highest weight.

For details, please see page 25, lines 291 to 293.

Comment 18: There a few punctuation and sentence structures errors. For instance, the last sentence under data collection needs to be qualified. Also avoid sub-headings in the discussion

Response: We have sought professional assistance to polish the language of the manuscript

Comment 19: "Based on the KAPB model, this study constructed a standardized training system for hospital infection prevention and control for new medical staff in the internal medicine ICU through literature research, qualitative interviews, expert interviews and AHP." The AHP has not been defined and not presented in the methods

Response: Thank you so much for pointing out our obvious mistake and we have added the definition of AHP in the manuscript.

The subjective judgments of the experts were subjected to quantitative scaling using the analytic hierarchy process [21]. The Saaty scale was determined based on the mean difference in importance assignments of evaluation indicators in the second round of expert consultation, and a hierarchical structure model was established. A judgment matrix was constructed, and single-level ranking and consistency checks were performed to determine the weight and composite weight of each index. A consistency ratio below 0.1 indicates a reasonable weight distribution for each indicator and good consistency [22].

For details, please see page 10, lines 182 to 189.

Comment 20: The authors should also discuss the limitations of the study

Response: We have added a description of the limitations of this study in the conclusion. 

Nonetheless, this study is not without limitations. The consulted experts hail solely from two provinces in China, and the initially constructed training system is still predominantly within the theoretical framework.

For details, please see page 27, lines 328 to 330.

---

## [Decision Letter · Decision Letter 1]

16 Oct 2023

PONE-D-23-09513R1Construction of a standardized training system for hospital infection prevention and control for new medical staff in internal medicine ICUs based on the Delphi methodPLOS ONE

Dear Dr. Tang,

Thank you for submitting your manuscript to PLOS ONE. After careful consideration, we feel that it has merit but does not fully meet PLOS ONE’s publication criteria as it currently stands. Therefore, we invite you to submit a revised version of the manuscript that addresses the points raised during the review process.

We look forward to receiving your revised manuscript.

Kind regards,

Arghya Das, MD

Academic Editor

PLOS ONE

Journal Requirements:

Reviewers' comments:

Reviewer's Responses to Questions

**Comments to the Author**

1. If the authors have adequately addressed your comments raised in a previous round of review and you feel that this manuscript is now acceptable for publication, you may indicate that here to bypass the “Comments to the Author” section, enter your conflict of interest statement in the “Confidential to Editor” section, and submit your "Accept" recommendation.

Reviewer #1: All comments have been addressed

Reviewer #2: All comments have been addressed

2. Is the manuscript technically sound, and do the data support the conclusions?

Reviewer #1: Yes

Reviewer #2: Yes

3. Has the statistical analysis been performed appropriately and rigorously? 

Reviewer #1: Yes

Reviewer #2: Yes

4. Have the authors made all data underlying the findings in their manuscript fully available?

Reviewer #1: Yes

Reviewer #2: Yes

5. Is the manuscript presented in an intelligible fashion and written in standard English?

Reviewer #1: Yes

Reviewer #2: Yes

6. Review Comments to the Author

Reviewer #1: Not applicable. The authors have met all the above listed criteria. There was no dual publication, research ethics, or publication ethics

Reviewer #2: The authors have significantly improved the content and structure of the manuscript and have addressed the comments appropriately.

Below are a few corrections required:

1. The authors can join the sentences in line 45 with a conjunction

2. Avoiding subheadings in the discussion is ideal

3. Line 327 seems to be incomplete.

4. The section on limitation could be inserted as the last paragraph in the discussion section

7. PLOS authors have the option to publish the peer review history of their article (what does this mean?). If published, this will include your full peer review and any attached files.

Reviewer #1: **Yes: **Victor Abiola Adepoju

Reviewer #2: No

---

## [Author Response · Author response to Decision Letter 1]

1 Nov 2023

Reviewer #2 (Comments to the Authors):

Comment 1: The authors can join the sentences in line 45 with a conjunction

Response: Thanks for your suggestion, we have joined the sentence in line 45 with a conjunction.

In conclusion, the proposed standardized training system for infection prevention and control among new medical staff in the internal medicine ICU is both scientifically sound and practical, which can contribute to improved patient safety, reduced healthcare costs, and enhanced overall quality of care in internal medicine ICUs.

For details, please see page 3, line 45.

Comment 2: Avoiding subheadings in the discussion is ideal

Response: Thank you very much for your valuable comments. We have removed the subheadings from the discussion.

Comment 3: Line 327 seems to be incomplete

Response: Thanks for your reminding, we have completed this sentence.

In addition, it can strengthen patient safety and reduce healthcare-associated infections.

For details, please see page 27, line 336.

Comment 4: The section on limitation could be inserted as the last paragraph in the discussion section

Response: Thanks for your suggestion, we have moved the section on limitations from the summary to the last part of the discussion. 

For details, please see page 27, lines 321 to 326.

---

## [Editor Report · Decision Letter 2]

6 Nov 2023

Construction of a standardized training system for hospital infection prevention and control for new medical staff in internal medicine ICUs based on the Delphi method

PONE-D-23-09513R2

Dear Dr. Tang,

We’re pleased to inform you that your manuscript has been judged scientifically suitable for publication and will be formally accepted for publication once it meets all outstanding technical requirements.

Kind regards,

Arghya Das, MD

Academic Editor

PLOS ONE
---

## [Editor Report · Acceptance letter]

9 Nov 2023

PONE-D-23-09513R2 

Construction of a standardized training system for hospital infection prevention and control for new medical staff in internal medicine ICUs based on the Delphi method 

Dear Dr. Tang:

I'm pleased to inform you that your manuscript has been deemed suitable for publication in PLOS ONE. Congratulations! Your manuscript is now with our production department. 

Kind regards, 

on behalf of

Dr. Arghya Das 

Academic Editor

PLOS ONE